# Pain-related fear of movement dynamics in individuals with and without low back pain participating in weightlifting and/or powerlifting training

**Bernard X. W. Liew**[1]*, **Josce Syrett**[1], **Paul Freeman**[1], **David W. Evans**[2]

**1** School of Sport, Rehabilitation and Exercise Sciences, University of Essex, Colchester, Essex, United Kingdom, **2** Centre of Precision Rehabilitation for Spinal Pain, School of Sport, Exercise and Rehabilitation Sciences, University of Birmingham, Edgbaston, Birmingham, United Kingdom

* bl19622@essex.ac.uk, liew_xwb@hotmail.com

**Data Availability Statement:** The data is enclosed in the supplemental zip file.

**Funding:** The author(s) received no specific funding for this work.

## Abstract

### Purpose

Pain-free adults in the general population have been shown to possess unhelpful beliefs that certain movements and postures are harmful to the spine, potentially reinforcing fear-avoidance behaviour. Whether such beliefs occur in individuals undertaking regular power-lifting (PL) and Olympic weightlifting (OWL) training is unclear.

### Methods

In a cross-sectional study design, 67 individuals who participate in OWL and PL training completed an online survey. Demographic characteristics, training history, and self-reported perceptions of harm, on the 40-item Photograph Series of Daily Activities shortened electronic version (PHODA-SeV), were collected. After removing collinear variables, 13 items were entered into a network analysis, in which the adjusted correlations between items, and the centrality indices of each item (i.e., the degree of connection with other symptoms in the network) were quantified.

### Results

Twenty-one (31.3%) participants had LBP symptoms. The pairwise correlations with the greatest magnitudes were between images of 'leg stretch' and 'jumping' (0.32 [95%CI 0.08 to 0.45]) and two images depicting ironing (0.32 [95%CI 0.05 to 0.54]) respectively. The three most Central (connected) items were 'stair ascend', 'walking with groceries', and 'mopping with spine flexion'.

### Conclusions

For individuals training in OWL and PL, images reflecting walking, rather than those depicting high spinal flexion angle, had greater connectivity to other activity items. In addition, the strongest correlations were not between items reflecting high spinal flexion angle. Future

**Competing interests:** The authors have declared that no competing interests exist.

studies that investigate the relationship between different intensities of OWL and PL training and the dynamics of pain-related fear are warranted.

## Introduction

Powerlifting (PL) and Olympic weightlifting (OWL) are amongst the most commonly practiced strength-based sports. The goal of both sports is to lift the maximum weight possible over a single repetition [1]. Low back pain (LBP) is one of the most common musculoskeletal pain disorders in both sports [1]. This is perhaps not surprising given that high spinal loads ($> 17,000$ N) can be experienced during these lifts [2]. Outside of the lifting sport populations, pain-related fear has been proposed as an important risk factor, prognostic factor, effect modifier, and mediator of treatments for LBP [3–5]. Pain-related fear has been investigated not only in people with LBP [3–6], but also in pain-free adults from the general population [7–9]. Whether the effect of repetitive training with high spinal-load activities confers a protective or harmful effect on pain-related fear is presently uncertain due to a dearth of studies in athletic cohorts.

The fear-avoidance model (FAM) has been one of the most influential theoretical models in the management of LBP. The FAM proposes that a painful episode could result in heightened pain-related fear, resulting ultimately in activity avoidance and greater disability [10, 11]. The most common type of pain-related fear to have been investigated is fear of movement (kinesiophobia), which is most commonly assessed using a version of the Tampa Scale for Kinesiophobia (TSK) self-reported questionnaire [12]. The summed score of such questionnaires is used as a 'latent' measure of the magnitude of kinesiophobia. A limitation of such fear evaluations is the lack of contextual specificity of the object of fear [13, 14]. In other words, it is not clear which particular movements or activities induce fear from the aggregate score of kinesiophobia questionnaires. The Photograph Series of Daily Activities (PHODA) [15], and more recently a shortened electronic version (PHODA-SeV) [13, 16], was created to overcome this by having participants rate their fear perceptions on images depicting various activities of daily living.

A key theoretical construct behind the use and interpretation of questionnaire-based methods is known as the 'reflective model' (RM) [17]—observed questionnaire item responses are determined by a latent trait. The RM model within the present context theorises that an individual possesses fear of movement, which results in them providing a higher score on an item within a kinesiophobia questionnaire. The RM was the theoretical approach adopted in the development of the PHODA-SeV, which reported that the questionnaire can be explained by a single factor structure [13]. The RM model in psychology is analogous to traditional disease models, which state that specific diseases result in the manifestations of symptoms [18]. The main limitation of the RM model in psychology is that, unlike in traditional diseases (e.g. cardiovascular disease) where the pathology can exist independent from the symptoms, such scenarios are very unlikely in mental health disorders (e.g. being depressed without feeling blue) [18]. In contrast to RM, another more recent theoretical approach towards understanding psychological states is the network approach [18]. In this, individual items on a questionnaire reflect different symptoms, and it is the interaction of these symptoms that gives rise to the psychological trait [18]. For example, in the network model, fear of movement emerges as a result of the simultaneous perception of fear in a number of activities.

In the only prior study of the PHODA-SeV on pain-free adults [8], the image of a person shovelling soil evoked the greatest magnitude of self-reported fear. By contrast, the image of a

person walking upstairs evoked the lowest magnitude of self-reported fear [8]. Interestingly, the image of a person falling backward was the only single-item on the PHODA-SeV that significantly correlated with an objective measure of lifting range of motion (ROM) [8]. In addition, the single item of lifting correlated highly with the item of shovelling (r = 0.80), whilst the former correlated moderately with falling (r = 0.46) [8]. A previous study on people with LBP reported that the fear rating of an image depicting a person lifting correlated with TSK scores by a magnitude of r = 0.68 [19]. A limitation of repeated univariate analysis of the individual PHODA-SeV items is that relationships between items are unknown. Importantly, multivariate relationships could reveal common triggers of fear (e.g. activities with high spinal flexion), and identify the most important object of fear from which other objects project to or from.

The current study aimed to explore a fear of movement profile in individuals who participate in OWL and PL sports. A previous study reported that the strongest correlation with lifting ROM were the items of falling backward and back twisting [8]. Similar to the conductance of network meta-analysis [20], we can posit that items reflecting falling backward and back twisting would be highly correlated with each other. Hence, we hypothesised that falling backwards and back twisting would have the strongest correlation in the present study. Second, we also hypothesised that images reflecting similar activity types (e.g. activities with high spinal flexion) would demonstrate the highest correlation with one another, compared to dissimilar activities (e.g. climbing stairs and getting out of bed). Third, we hypothesised that the image depicting a person shovelling soil would be the most important (i.e. connected) item in the network. Lastly, we hypothesised that images depicting high spinal flexion angle would be amongst the top three most connected items in the network.

## Methods

### Study design

A cross-sectional study involving a single online survey was undertaken between October 2021 to February 2022. The study received ethical approval from the University of Essex Human Ethics Committee (ETH2122-0110) and was conducted in accordance with the Declaration of Helsinki. Written informed consent was provided by participants before their inclusion.

### Sample size

The present study's sample size was calculated based on a broader project investigating fear of movement and LBP beliefs in individuals participating in OWL and PL training. The power calculation was based on a one-sample t test where we hypothesised that participants in this cohort would have a lower PHODA-SeV total score compared to the general population. A power calculation was performed using the *pwr* package in R statistical software, based on a one-sample t-test. A previous study reported an improvement in the aggregate PHODA-SeV score after graded activity treatment by 10/100 [13]. Based on a standard deviation (SD) of 12/100 [8], this improvement would translate to an effect size of 0.8, which we hypothesised to be the effect size of the difference between participants in this cohort and the general population. To detect a more conservative, moderate effect size of 0.5, 64 participants were required to provide a statistical power of 0.8, at an alpha of 0.05. To account for dropouts, 70 participants were recruited.

Performing an a priori sample size calculation for a network analysis is extremely challenging given that for data-driven methods, there is no a priori single hypothesis [21, 22]. However, given a preliminary network, a post-hoc sample size estimation can be performed, based upon achieving a desired level of stability in the network findings [22]. This is akin to determining

sample size to achieve a desired precision of effect size, rather than the magnitude of effect size [23].

## Participants

Participants were eligible if they were: 1) aged between 18 to 60 years old, 2) self-reported competing and/or training in OWL and PL, and 3) had an adequate command of the English language sufficient to complete the survey.

## Questionnaire

The electronic online survey was hosted on Qualtrics software (Qualtrics, Provo, UT). The following descriptive characteristics were collected from all participants: age, gender at birth, height, body mass, training sports type, years of training, frequency of training, current low back pain intensity (on a 0–100 visual analogue scale where 0 = no pain and 100 = maximal pain), maximal training back squat, deadlift and bench press load as self-reported.

The PHODA-SeV is a 40-image item questionnaire (Table 1), which is freely available to download (https://ppw.kuleuven.be/ogp/software/phodasev-en.zip) and quick to administer. All 40 images can be found in the included supplementary material. Each image is scored using a 100-point visual analogue sliding scale from 0 (not harmful at all) to 100 (extremely harmful). In a RM, the total score is obtained by averaging over all 40 item responses [13]. The PHODA-SeV has demonstrated high internal consistency (Cronbach alpha > 0.75) and excellent test-retest reliability (ICC > 0.90) [13, 16]. The presented image sequence was not randomised.

## Approach to network analysis

**Software and packages.** The dataset was analysed with R statistical software (version 4.1.2). Several packages were used to perform the analyses, including *qgraph* [24] for network estimation, and *bootnet* [22] for stability analysis and post-hoc sample size estimation.

**Variables included in network analysis.** Moderately high collinear items were removed using a threshold $\geq 0.6$ [25], which resulted in 13 items being retained for the network analysis. A threshold of 0.6 was selected as a higher threshold would have left too few items to benefit from performing a multivariate technique such as network analysis. We treated the items of the PHODA-SeV as continuous variables and applied a nonparanormal transformation to ensure these variables were multivariate normally distributed [26]. A network structure is composed of nodes (variables influencing each other) and edges (connections or associations between nodes). Each edge in the network represents either a positive regularised association (blue edges) or a negative regularised association (red edges). A negative association between items indicates that an increase in the magnitude of one item results in a decrease in the magnitude of the second item. A positive association between items indicates that an increase in the magnitude of one item results in an increase in the magnitude of the second item. The thickness and colour saturation of an edge denotes the strength of the association between two nodes, which is akin to the magnitude of a Pearson correlation coefficient value.

**Network estimation.** When estimating the network, the graphical least absolute shrinkage and selection operator (LASSO) regularization LASSO [27] was used to elicit a sparse model. If $S$ represents the sample variance–covariance matrix, LASSO aims to estimate $K$ by maximizing

**Table 1. Description of items of the PHODA-SeV [13].**

| Item | Original PHODA number | Description |
|------|----------------------|-------------|
| Q1 | 100 | Drilling a hole in a stone wall above the head |
| Q2 | 11 | Taking book from shelf behind oneself (with twisted back) |
| Q3 | 14 | Ironing in standing position |
| Q4 | 15 | Ironing in sitting position |
| Q5 | 18 | Lifting a filled basket while walking up the stairs |
| Q6 | 20 | Lifting beer crate out of car with slightly bent back |
| Q7 | 22 | Carrying a shopping bag with one hand while walking |
| Q8 | 23 | Carrying two shopping bags with both hands while walking |
| Q9 | 26 | Carrying rubbish bag with one hand while walking |
| Q10 | 27 | Clearing out the dishwasher with bent back |
| Q11 | 28 | Taking a box from the sink cupboard above the head |
| Q12 | 29 | Vacuum cleaning under coffee table with bent back |
| Q13 | 33 | Mopping floor with a squeegee with slightly bent back |
| Q14 | 36 | Leg stretch exercise on a fitness device |
| Q15 | 40 | Back twist exercise on a fitness device |
| Q16 | 44 | Back muscle exercise bending forward on a fitness device |
| Q17 | 47 | Taking a box filled with bottles from a shelf above the head |
| Q18 | 49 | Trampoline jumping |
| Q19 | 50 | Rope skipping |
| Q20 | 51 | Abdominal muscle exercises on the floor with fitness device |
| Q21 | 57 | Making the bed with bent back |
| Q22 | 59 | Getting out of bed by first placing one foot on the ground |
| Q23 | 60 | Walking up the stairs |
| Q24 | 61 | Walking down the stairs |
| Q25 | 73 | Cleaning the windows with arm stretched above the head |
| Q26 | 74 | Riding a bicycle in a street with speed bumps |
| Q27 | 8 | Picking up shoes from floor squatting down |
| Q28 | 83 | Lifting a toddler (1–2 y) from its cot with bent back |
| Q29 | 85 | Carrying a child (5 y) on the hip |
| Q30 | 92 | Doing the dishes in standing position |
| Q31 | 93 | Running through the forest |
| Q32 | 94 | Walking through the forest |
| Q33 | 95 | Cycling from a low kerb |
| Q34 | 96 | Looking aside while cycling |
| Q35 | 98 | Falling backward on the grass |
| Q36 | 99 | Mowing the lawn manually |
| Q37 | 2 | Shoveling soil with bent back |
| Q38 | 3 | Lifting flowerpot squatting down |
| Q39 | 4 | Lifting flowerpot with slightly bent back |
| Q40 | 7 | Picking up shoes from floor with bent back |

the penalised likelihood function:

$$log \det(K) - trace\ (SK) - \lambda \sum_{<i,j>} |\kappa_{ij}|$$

A sparse model, compared to a saturated model, results in easier interpretability given that the former has comparatively fewer edges to explain the covariation structure of the data [21].

The LASSO uses a tuning parameter to control the sparsity of the network, which was selected by minimizing the Extended Bayesian Information Criterion (EBIC) [28]. The graphical LASSO was run for 100 values of the $\lambda$ logarithmically spaced between the maximal value of the tuning parameter at which all edges are zero ($\lambda_{max}$), and $\lambda_{max}/100$. For each of these graphs, the EBIC is computed and the graph with the lowest EBIC is selected [21, 29].

**Node centrality.** Centrality indices provide a measure of a node's importance (i.e. connectedness), and are based on the pattern of connectivity of a node of interest with its surrounding nodes–with the ensuing information potentially useful for guiding future interventions [30].

In the present study, we calculated three centrality indices:

- Strength centrality, defined as the sum of the weights of the edges (in absolute value) incident to the node of interest [31, 32]. Clinically, a high Strength node represents a logical and efficient therapeutic target, because changing this node has a strong, direct and quick influence on other nodes within the network.

- Closeness centrality [31], defined as the inverse of the sum of the length of the shortest paths between a node of interest and all other nodes in the network. Clinically, a high Closeness node may represent a potentially good therapeutic target, because the effects changing this node will spread more quickly throughout the network, via direct and indirect connections.

- Betweenness centrality, defined as the number of times a node acts as a bridge along the shortest path between two other nodes [31, 33]. Clinically, a high Betweenness node may suggest that the node represents a potential mediator since it acts as a bridge for connecting different nodes.

**Accuracy and stability.** We assessed the accuracy of the edge weights and the stability of three centrality indices using bootstrapping [22], which re-estimates the network parameters several times using a resampling technique. Accuracy and stability analyses are essential in network analysis studies to correctly interpret the results obtained. We bootstrapped using 1000 iterations, to generate 95% confidence intervals (CI) of all edge weights.

To gain an estimate of the variability of the three centrality indices, we applied the case-dropping subset bootstrap [22]. This procedure drops a percentage of participants, re-estimates the network and re-calculates the three centrality indices; producing a centrality-stability coefficient (CS-coefficient). CS reflects the maximum proportion of cases that can be dropped, such that with 95% probability the correlation between the centrality value of the bootstrapped sample versus that of the original data, would reach a certain value, taken to be a correlation magnitude of 0.7 presently. It is suggested that $CS_{cor = 0.7}$ should not be below 0.25 and better if greater than 0.5 [22].

**Post-hoc sample size analysis.** Given the network estimated, the *netSimulator* function of the *bootnet* package was used to calculate six performance indices: 1) correlation between the edge weights; 2) sensitivity–the proportion of edges present, 3) specificity–proportion of missing edges; correlation between 4) Strength, 5) Closeness, and 6) Betweenness centrality measures of the given network against that of the re-estimated network using different sample sizes [21]. The *netSimulator* function simulates data based on a priori network structure and edge weights, similar to how a standard statistical power simulation study simulates new data based on an hypothesised effect size being detected. Using our original network structure and edge weights, new data with varying sample sizes were simulated. Herein, we varied the sample size, n, across these values: 50, 100, 150, 200, 250, and 500. For each sample size, 2000 bootstrap

samples with replacement were performed to re-estimate the network and calculate the six performance indices.

## Results

Descriptive characteristics of participants can be found in Table 2. 67 participants were included in the analysis, as three participants had missing PHODA item scores. The mean (standard deviation [SD]) reported for all 40 items of the PHODA-SeV (original scale) are displayed in Fig 1. Fig 2A shows the network, while Fig 2B shows the network with the images embedded as nodes. Edge weights and variability (Fig 3, S1 Table), and centrality indices (Fig 4) values are reported graphically in the manuscript.

The five edges with greatest weight magnitudes were between Q14-Q18 (leg stretch and jumping) (0.32 [95%CI 0.08 to 0.45]), Q3-Q4 (two images depicting ironing) (0.32 [95%CI 0.05 to 0.54]), Q2-Q35 (twisting and falling backward) (0.28 [95%CI 0.08 to 0.41]), Q4-Q30 (ironing and dishwashing) (0.27 [95%CI 0.00 to 0.46]), and Q23-Q32 (climbing stairs and walking) (0.27 [95%CI 0.04 to 0.50]) (Fig 2).

The stability of the centrality measures, $CS_{cor = 0.7}$, of Strength, Closeness, and Betweenness were 0.134, 0.045, and 0 respectively. Given the instability of Closeness and Betweenness measures, we only report the Strength measure in the manuscript–which although lower than the

**Table 2. Descriptive characteristics of sample.**

| Variable | N = 67[1] |
|---|---|
| **Age (yrs)** | 27.70 (8.35) |
| **Gender** | |
| Female | 32 / 67 (48%) |
| Male | 35 / 67 (52%) |
| **Height (m)** | 1.71 (0.09) |
| **Body mass (kg)** | 74.26 (12.22) |
| **Training sport** | |
| OWL | 20 / 67 (30%) |
| PL | 25 / 67 (37%) |
| Both | 22 / 67 (33%) |
| **Years of training** | |
| 1–2 | 39 / 67 (58%) |
| 3–4 | 28 / 67 (42%) |
| **Frequency of training (days/week)** | |
| 2 | 1 / 67 (1.5%) |
| 3 | 6 / 67 (9.0%) |
| 4 | 12 / 67 (18%) |
| 5 | 34 / 67 (51%) |
| 6 | 9 / 67 (13%) |
| 7 | 5 / 67 (7.5%) |
| **Participants with LBP symptoms** | 21 (31.3%) |
| **LBP intensity (0–100)** | 12.13 (16.19) |
| **Back Squat (kg)** | 170.83 (54.61) |
| **Deadlift (kg)** | 201.59 (55.65) |
| **Bench Press (kg)** | 104.75 (36.04) |

[1]Mean (SD); n / N (%)

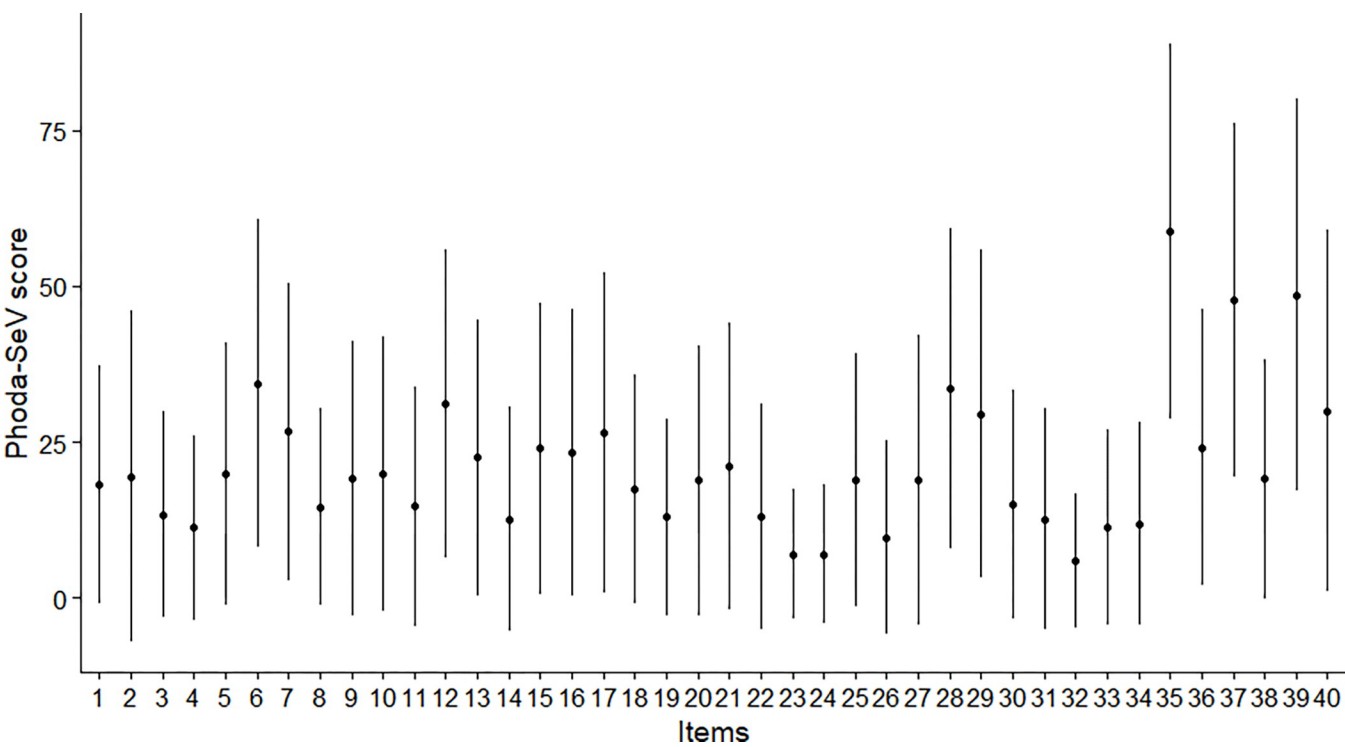

**Fig 1. Mean with error bars as one standard deviation of the individual item scores of the PHODA-SeV.**

recommended stability threshold of 0.25, still reflects the most stable measure presently. The top three items with the greatest Strength measures were Q23 (stair ascend) at 1.00, Q8 (walking with groceries) at 0.93, and Q13 (mopping with spine flexion) at 0.90 (Figs 2 and 4).

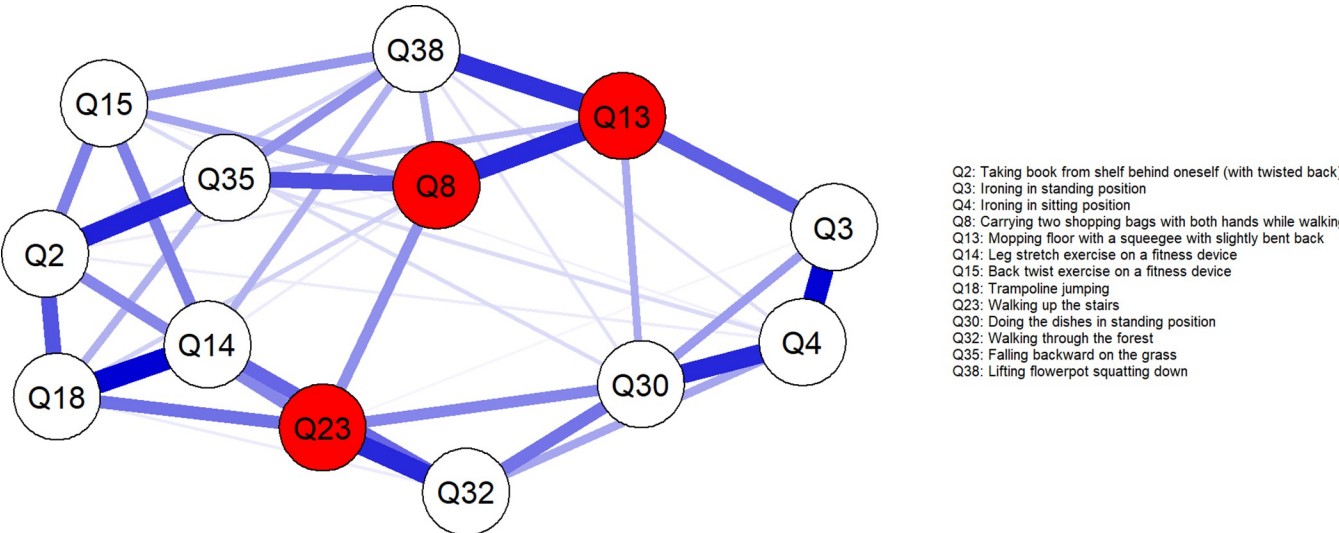

Q2: Taking book from shelf behind oneself (with twisted back)
Q3: Ironing in standing position
Q4: Ironing in sitting position
Q8: Carrying two shopping bags with both hands while walking
Q13: Mopping floor with a squeegee with slightly bent back
Q14: Leg stretch exercise on a fitness device
Q15: Back twist exercise on a fitness device
Q18: Trampoline jumping
Q23: Walking up the stairs
Q30: Doing the dishes in standing position
Q32: Walking through the forest
Q35: Falling backward on the grass
Q38: Lifting flowerpot squatting down

**Fig 2. Network analysis of the association between items of the PHODA-SeV.** (a) nodes with characters as item identifier, (b) nodes with images as item identifier. Edges represent connections between two nodes and are interpreted as the existence of an association between two nodes, adjusted for all other nodes. Each edge in the network represents either positive regularised adjusted associations (blue edges) or negative regularised adjusted associations (red edges). The thickness and colour saturation of an edge denotes its weight (the strength of the association between two nodes). Nodes colored red are those with a high Strength centrality measure.

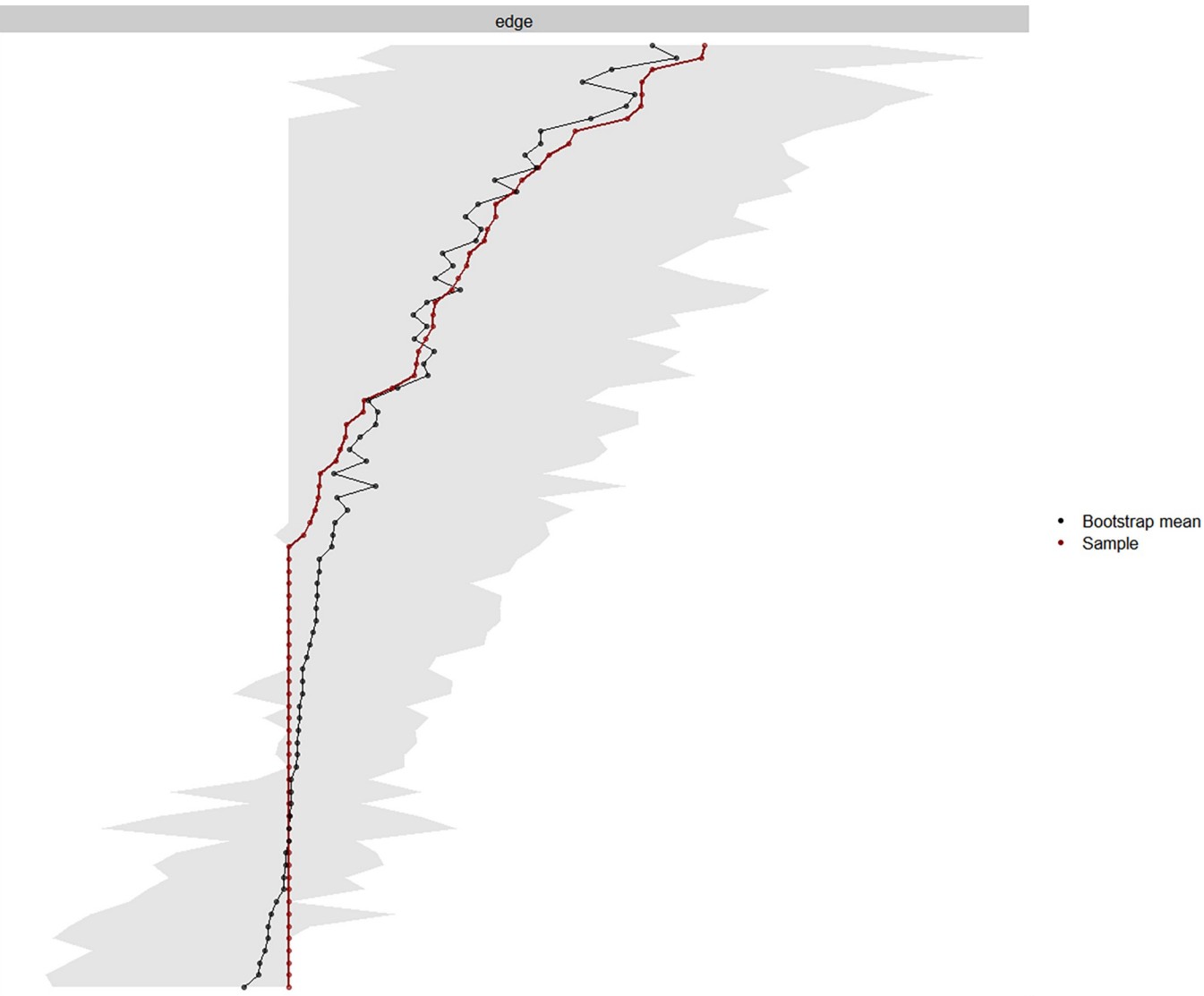

**Fig 3. Bootstrapped 95% confidence interval of the estimated edge weights of the network.** 'Bootstrap mean' represents the mean value of edge weights across the bootstrapped samples. 'Sample' represents the magnitude of edge weights of the original network built on the original dataset.

From Fig 5, the correlation of the estimated edge weights and sensitivity increased significantly from n = 100 to n = 150, suggesting that assuming our estimated was true, a sample size of at least n = 150 is required. The correlation between the presently estimated Centrality indices and the re-estimated indices with different sample sizes also increased significantly from n = 100 to n = 150, but only when n = 250 did the median correlation exceed a threshold of 0.7 (Fig 5).

## Discussion

Pain-related fear is thought to be an important construct underpinning the management of a complex disorder like LBP and disability attributed to it. The PHODA-SeV has been used to understand fear of movement in people with and without LBP, but not in individuals who routinely engage in high-spinal loading sports. The greatest correlation was between Q14-Q18

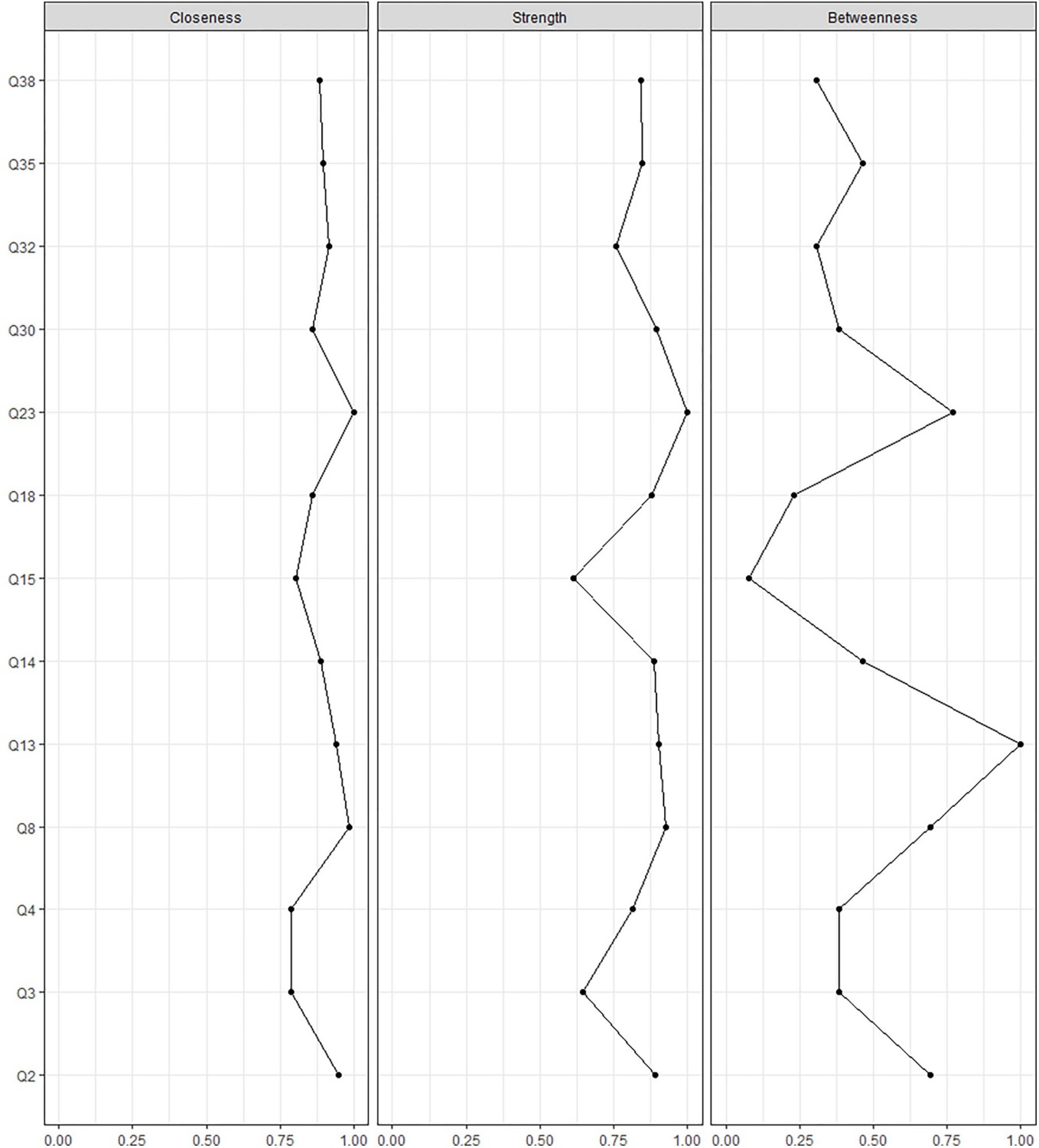

**Fig 4. Centrality measures of Closeness, Strength, and Betweenness of each node in the network.** Centrality value of 0 indicates no importance and 1 indicates maximal importance.

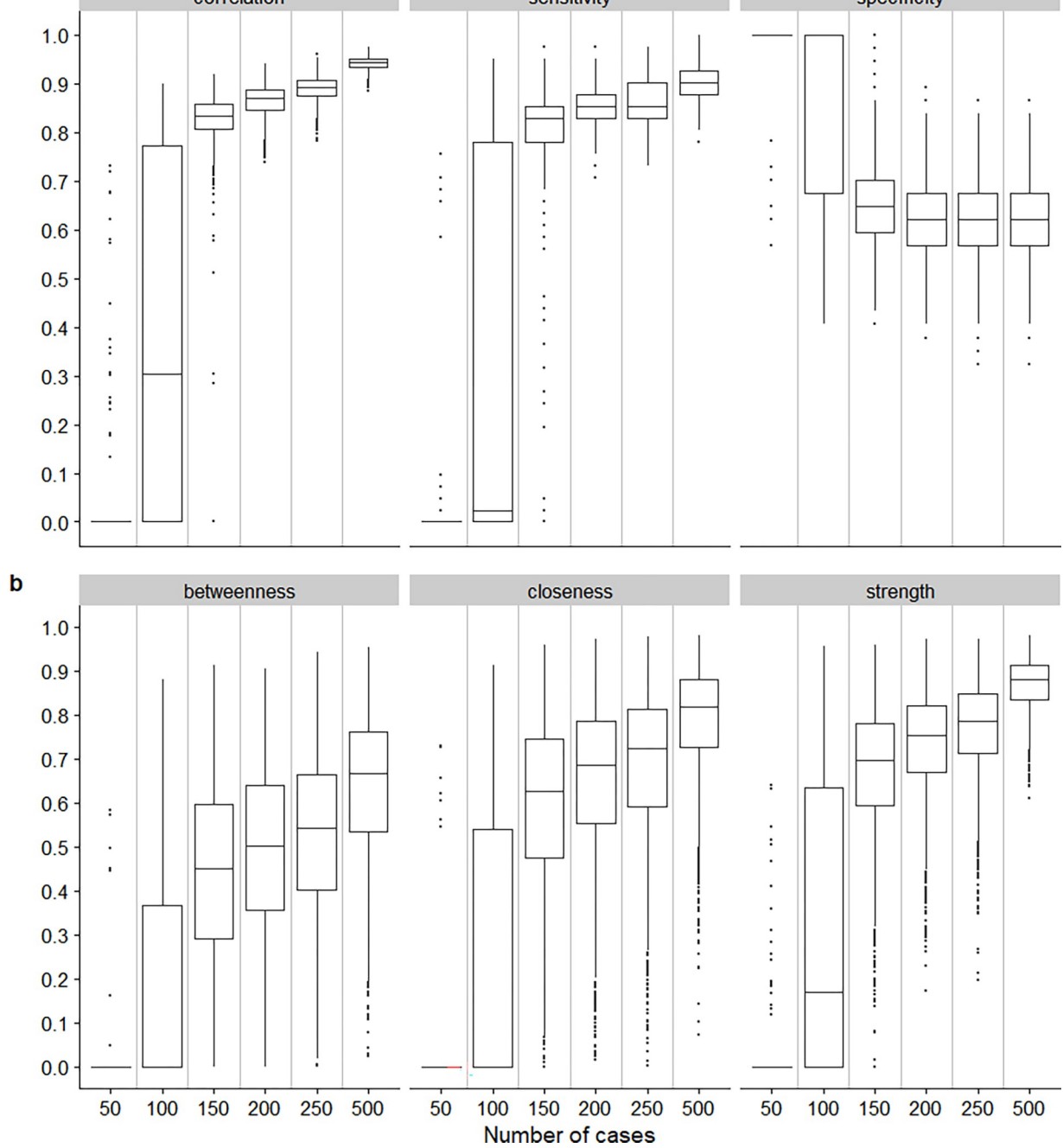

**Fig 5.** Boxplots of the estimated a) correlation of edge weights, sensitivity, specificity, and b) correlation of the Centrality measures between the given network and the re-estimated network with various sample size.

(leg stretch and jumping) and the third strongest correlation was between Q2-Q35 (twisting and falling backward)–which supported our first hypothesis. In support of our second hypothesis, similar activities (e.g. Q33-Q34, two cycling images) were more highly correlated than dissimilar activities (e.g. Q22-Q23, getting out of bed and climbing stairs). In contrast to our third and fourth hypotheses, the most important items in the network were climbing stairs, walking with groceries and mopping the floor, and not shovelling soil or high-spinal flexion angle activities.

Items with the highest correlations depicted similar activity postures and activity types. For example, Q14-Q18 (leg stretch and jumping) reflect the behaviours of a squat motion, Q3-Q4-Q30 (two ironing images and dishwashing) were items where spinal flexion angle magnitude were visually similar, Q2-Q35 (twisting and falling backward) have elements of spinal axial rotation, and Q23-Q32 (climbing stairs and walking) relate to walking gait. One of the most powerful aspects of network analysis is that the visualisation reveals how fear on specific activities may (in)directly cause or reflect a consequence, of fear on other activities. Our findings are congruent with qualitative research that reported how some LBP sufferers view their activity limitation as a consequence of other preceding limitations in activity [34]. We thus view network analysis as a powerful quantitative tool that complements qualitative research in unravelling the complexity of pain-related fear.

Previous studies reported that shovelling soil, falling backwards, and lifting a pot with a bent back were the top three fear-evoking items in people with and without LBP [8, 16]. Two out of these three items are associated with high-spinal flexion angles [8, 16]. Lay people with LBP reported a fear level between 76 to 85/100 on these three items, whilst people without LBP reported a level between 47 to 52/100 [8, 16]. A limitation of these two previous studies was that the importance of these three highest scored items relative to other items was not quantified. Interestingly, despite training and competing in OWL and PL, the item depicting mopping the floor, an image of a woman with a high spinal flexion angle, was in the top three most Central items. Our findings contrast with a previous study which reported that athletes reported lower frequencies of fear avoidance behaviour in the social and physical domain compared to the non-athletes [35]. Differences in pain-related fear between trained and less-trained individuals may vary depending on the specificity of the questionnaire used to assess fear.

We predicted that activities with a high spinal flexion angle would demonstrate the highest correlations with other activity types. We based this on a previous finding that a single PHODA-SeV lifting image demonstrated a correlation of 0.81 with the total PHODA score [8]. In addition, fear of movement has routinely been studied within the context of activities involving high spinal flexion angle [9, 36], suggesting that fear of spinal flexion appears to be either a common cause, effect, or mediator that drives other objects of fear. In contrast to our predictions, locomotion activities (Q8, Q23) had the greatest Strength centrality score. Items with high Strength have a strong, direct, and quick influence on other items within the network. It may be that regular participation in OWL and PL prevented items with high spinal flexion angles from having the greatest Strength score. Future studies comparing Centrality scores of the network items, between those who regularly train in OWL and PL and those who do not, would help understanding of the importance of different activity participation on fear dynamics.

An interesting observation was that items with high average scores (e.g. Q35, falling backwards) were not the most Central items. There is consistency across literature and the present study that individuals report high fear levels of falling backward [8, 16]. This is not surprising given that a fall may lead to serious injuries, including triggering a LBP episode. However, the lack of Centrality importance to an image of falling could imply that participants rate this scenario as unlikely. In contrast, lower-scoring items of walking and stair climbing (Q8, Q23) had high Centrality. This could be due to individuals recognising the potential higher likelihood that these activities would be frequently affected by LBP during daily living.

The presented network visualisation (Fig 2) is both clinically intuitive, and can offer unique clinical insights which may guide the management of LBP. First, knowing the item pairs with strong associations may provide an indirect, and potentially easier, path toward reducing fear of specific activities. For example, Q14-Q18 (leg stretch and jumping) was the item pair with

the strongest association. This could mean that strategies to reduce fear associated with performing a leg stretch manoeuvre could also be used to reduce the fear associated with jumping. Second, it may often be desirable, especially in a busy clinical environment, to identify a smaller set of therapeutic targets that can maximally benefit other factors not directly treated. Herein, items in the network with the largest Centrality values represent candidate therapeutic targets, because they are most connected to other items.

This study is not without limitations. First, a cross-sectional study cannot differentiate if two variables covary because of covariation in means between-subjects, or because of covariation between changes from the mean (i.e. within-subject) [37]. Accordingly, extrapolating our findings to longitudinal changes over time within a participant should be done with caution. Second, it could be argued that because the sequence of item presentation was not randomised in the present study, items presented in close sequence were more likely to be scored similarly because of recall bias. This could have affected the correlation of some items, such as between Q3 and Q4, which both depict images of a woman ironing. Whilst this is a limitation in the present study, we also note that only one item pair (Q3-Q4) was in our five greatest correlations. Third, we did not record the duration of LBP symptoms from our participants. A previous review reported that in patients where LBP duration was less than six months, fear was more likely to be associated with disability than in those where the duration was more than six months [5]. This suggests that the dynamics of fear may differ in individuals with different LBP duration. Examining if the present findings are consistent across clinical LBP subgroups should be explored in future research. Lastly, as evidenced by our post-hoc analysis, our sample size was too low to achieve stability in our Centrality findings. Our present findings will enable future researchers to perform a priori sample size calculations for the network analysis of PHODA-SeV.

## Conclusions

Our network analysis of the PHODA-SeV questionnaire in a cohort of individuals training in OWL and PL revealed that items reflecting walking had greater direct links to other activity items (i.e. Strength centrality), and not items that involved high-spinal flexion angle activities. In addition, the strongest correlations were not between items reflecting high spinal flexion angle. This lack of a relationship could reflect the potential influence of OWL and PL on the attenuation of the association between activities involving high spinal flexion angle and other activity types. Future studies that compare the networks of the PHODA-SeV between different cohorts would be useful to understand how frequent OWL and PL training influences the associations and centrality of different activity items.

## Supporting information

**S1 Table. Partial correlation values and 95% confidence interval.**
(DOCX)

**S1 Data.**
(CSV)

## Author Contributions

**Conceptualization:** Bernard X. W. Liew, Josce Syrett, David W. Evans.

**Data curation:** Josce Syrett.

**Formal analysis:** Bernard X. W. Liew, Josce Syrett.

**Investigation:** Josce Syrett, Paul Freeman.

**Methodology:** Bernard X. W. Liew.

**Project administration:** Bernard X. W. Liew, Paul Freeman, David W. Evans.

**Supervision:** Bernard X. W. Liew, Paul Freeman, David W. Evans.

**Validation:** Paul Freeman, David W. Evans.

**Visualization:** Bernard X. W. Liew, David W. Evans.

**Writing – original draft:** Bernard X. W. Liew, Josce Syrett, Paul Freeman, David W. Evans.

**Writing – review & editing:** Bernard X. W. Liew, Josce Syrett, Paul Freeman, David W. Evans.

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
