## [Decision Letter · Decision Letter 0]

22 Aug 2022

PONE-D-22-19501Pain-related fear of movement dynamics in athletes who train in weightlifting and/or powerliftingPLOS ONE

Dear Dr. Bernard X W Liew,

Thank you for submitting your manuscript to PLOS ONE. After careful consideration, we feel that it has merit but does not fully meet PLOS ONE’s publication criteria as it currently stands. Therefore, we invite you to submit a revised version of the manuscript that addresses the points raised during the review process.

Please submit your revised manuscript by Oct 06 2022 11:59PM. If you need more time than this to complete your revisions, please reply to this message or contact the journal office at plosone@plos.org. Please include the following items when submitting your revised manuscript:A rebuttal letter that responds to each point raised by the academic editor and reviewer(s). You should upload this letter as a separate file labeled 'Response to Reviewers'.A marked-up copy of your manuscript that highlights changes made to the original version. You should upload this as a separate file labeled 'Revised Manuscript with Track Changes'.An unmarked version of your revised paper without tracked changes. You should upload this as a separate file labeled 'Manuscript'.If applicable, we recommend that you deposit your laboratory protocols in protocols.io to enhance the reproducibility of your results. Protocols.io assigns your protocol its own identifier (DOI) so that it can be cited independently in the future. For instructions see: https://journals.plos.org/plosone/s/submission-guidelines#loc-laboratory-protocols. Additionally, PLOS ONE offers an option for publishing peer-reviewed Lab Protocol articles, which describe protocols hosted on protocols.io. Read more information on sharing protocols at https://plos.org/protocols?utm_medium=editorial-email&utm_source=authorletters&utm_campaign=protocols.

We look forward to receiving your revised manuscript.

Kind regards,

José M. Muyor

Academic Editor

PLOS ONE

Journal Requirements:

4. We note that Figure 2 and S1 in your submission contain copyrighted images. All PLOS content is published under the Creative Commons Attribution License (CC BY 4.0), which means that the manuscript, images, and Supporting Information files will be freely available online, and any third party is permitted to access, download, copy, distribute, and use these materials in any way, even commercially, with proper attribution. For more information, see our copyright guidelines: http://journals.plos.org/plosone/s/licenses-and-copyright.

a. You may seek permission from the original copyright holder of Figure 2 and S1 to publish the content specifically under the CC BY 4.0 license. 

Reviewers' comments:

Reviewer's Responses to Questions

**Comments to the Author**

1. Is the manuscript technically sound, and do the data support the conclusions?

Reviewer #1: Yes

Reviewer #2: Yes

2. Has the statistical analysis been performed appropriately and rigorously? 

Reviewer #1: Yes

Reviewer #2: Yes

3. Have the authors made all data underlying the findings in their manuscript fully available?

Reviewer #1: Yes

Reviewer #2: No

4. Is the manuscript presented in an intelligible fashion and written in standard English?

Reviewer #1: Yes

Reviewer #2: Yes

5. Review Comments to the Author

Reviewer #1: To the authors

This is an interesting and well-conducted study that explored the pain-related fear (using PHODA-SeV) in Olympic weightlifting and/or Powerlifting athletes. I suggested some changes that may improve the manuscript.

Minor comments

TITLE

Page 1, line 1. The title could have also included ‘athletes with low back pain’.

ABSTRACT

Page 2, line 22. The authors should describe the sample size and that athlete presented low back pain.

METHODS

Page 7, line 125. The range of age considered in the eligibility criteria seems wide considering that the subjects were athletes. It may be a limitation of the study.

RESULTS

Page 11, line 207. The supplementary Table S1 was not available for download.

Table 1. I suggest reporting the years of practice in ranges of periods (e.g., less than 1, between 1 and 2, etc).

Table 1. Duration of symptoms is associated with fear avoidance beliefs. The duration of low back pain should be described in table 1.

DISCUSSION

The clinical implication should be described in the discussion section.

The findings might be different if patients presented worse (higher level of pain intensity) or longer symptoms. It should be also discussed.

Reviewer #2: Dear Editor,

Thank you for allowing me to review the article entitled "Pain-related fear of movement dynamics in athletes who train in weightlifting and/or powerlifting". The study analyzes the fear of movement-related pain in athletes through a network analysis of the 40-item PHODA-SeV questionnaire. The manuscript provides an adequate review of previous studies, theoretical background and exposition of methodology and results achieved. The conclusions may be of interest to the readers of the journal and for this reason I recommend its publication after the modification of certain minors concerns that could improve the reading and comprehension of the text:

1. In the Introduction, the difference between the reflective model and network analysis is explained. However, given that the authors are committed to the network paradigm, some comparison of the results between the reflective model used in the original development of the questionnaire and the results obtained with the alternative model would be desirable in the discussion of the manuscript. (p. 5).

2. In the Method, although a power analysis is established for the calculation of statistical power, it still seems too small a number of participants to assess the validity of a questionnaire with 40 items, even to reach a moderate effect size of 0.5. What kind of parameters were used to obtain this calculation and what software was used to perform it (p. 7).

3. I suggest to further develop the criteria for excluding 70% of the items in the network analysis, reducing it to 13 items. (p. 8).

4. Give more information and references on how this regularized association index is calculated. I also suggest defining how to obtain the weights in gradients and color for better interpretation, for example, what indicates a positive or negative association. (p. 9).

5. Wouldn't a larger sample size be needed for the bootstraap procedure to be reliable? Please provide references to support that this method is well employed (p. 10).

6. Give more information on how the procedure for estimating the indices by varying the sample sizes per simulation was relaized. Please explain in more detail (p. 11).

7. In Table 1, it is not clear when using means and SD since it does not agree that there are no decimals in the last four variables.

8. Give the quantitative values of the items with more Strength. (p. 13).

 **********

---

## [Author Response · Author response to Decision Letter 0]

13 Sep 2022

Please see attached response file for properly formatted responses.

To both reviewers, Figure 2 was modified to remove the images of the PHODA as it requires a permissions statement from the owner of the images. I have not been able to secure the statement from the owner. I have replaced the images with a description taken from the original publication of the PHODA-SeV.

Comments are in bold, response in normal typeset, and excerpts from manuscript are in italics.

Reviewer #1 

This is an interesting and well-conducted study that explored the pain-related fear (using PHODA-SeV) in Olympic weightlifting and/or Powerlifting athletes. I suggested some changes that may improve the manuscript.

Reply: We thank the Reviewer for the positive comment and will address all comments below.

Minor comments

TITLE

Page 1, line 1. The title could have also included ‘athletes with low back pain’.

Reply: We have renamed the title to be:

Pain-related fear of movement dynamics in individuals with and without low back pain participating in weightlifting and/or powerlifting training

ABSTRACT

Page 2, line 22. The authors should describe the sample size and that athlete presented low back pain.

Reply: We have included these two items in the abstract.

Methods: In a cross-sectional study design, 67 individuals who participate in OWL and PL training completed an online survey. Demographic characteristics, training history, and self-reported perceptions of harm, on the 40-item Photograph Series of Daily Activities shortened electronic version (PHODA-SeV), were collected. After removing collinear variables, 13 items were entered into a network analysis, in which the adjusted correlations between items, and the centrality indices of each item (i.e., the degree of connection with other symptoms in the network) were quantified.

Results: Twenty-one (31.3%) participants had LBP symptoms. The pairwise correlations with the greatest magnitudes were between images of ‘leg press’ and ‘jumping’ (0.32 [95%CI 0.08 to 0.45]) and two images depicting ironing (0.32 [95%CI 0.05 to 0.54]) respectively. The three most Central (connected) items were ‘stair ascend’, ‘walking with groceries’, and ‘mopping with spine flexion.

METHODS

Page 7, line 125. The range of age considered in the eligibility criteria seems wide considering that the subjects were athletes. It may be a limitation of the study.

Reply: We agree with the Reviewer that our cohort was more reflective of people who trained using OWL and PL, rather than athletes. Hence, we have reworded all instances of the term “athletes” in the manuscript.

RESULTS

Page 11, line 207. The supplementary Table S1 was not available for download.

Reply: We are unclear as to the reason for this. We have ensured that the supplementary content can be downloaded from the pdf proof, unzipped, and all files opened. We will check with the editorial team to ensure the supplementary files can be opened in the resubmission process.

Table 1. I suggest reporting the years of practice in ranges of periods (e.g., less than 1, between 1 and 2, etc).

Reply: We have changed the table to be:

Years of training 

1-2 39 / 67 (58%)

3-4 28 / 67 (42%)

Table 1. Duration of symptoms is associated with fear avoidance beliefs. The duration of low back pain should be described in table 1.

Reply: We thank the Reviewer for this important comment. Unfortunately, we did not record the duration of LBP from the participants. We have added this in the limitation in L347:

Third, we did not record the duration of LBP symptoms from our participants. A previous review reported that in patients where LBP duration was less than six months, fear was more likely to be associated with disability than in those where the duration was more than six months [5]. This suggests that the dynamics of fear may differ in individuals with different LBP duration.

DISCUSSION

The clinical implication should be described in the discussion section.

Reply: We have added a paragraph in the Discussion L328:

The presented network visualisation (Figure 2) is both clinically intuitive, and can offer unique clinical insights which may guide the management of LBP. First, knowing the item pairs with strong associations may provide an indirect, and potentially easier, path toward reducing fear of specific activities. For example, Q14-Q18 (leg press and jumping) was the item pair with the strongest association. This could mean that strategies to reduce fear associated with performing a leg press machine could also be used to reduce the fear associated with jumping. Second, it may often be desirable, especially in a busy clinical environment, to identify a smaller set of therapeutic targets that can maximally benefit other factors not directly treated. Herein, items in the network with the largest Centrality values represent candidate therapeutic targets, because they are most connected to other items.

The findings might be different if patients presented worse (higher level of pain intensity) or longer symptoms. It should be also discussed.

Reply: We thank the Reviewer for this comment. We agree that if our cohort had a greater sample size which would allow subgrouping of patients with varying pain intensity and duration, it will enable a comparison of fear networks between subgroups. However, our sample size precluded us from performing such an analysis. Hence, a thorough discussion on whether distinct LBP subgroups would present with the different fear networks would be very speculative. 

We have added the following to the limitation section:

Examining if the present findings are consistent across clinical LBP subgroups should be explored in future research.

Reviewer #2

 Thank you for allowing me to review the article entitled "Pain-related fear of movement dynamics in athletes who train in weightlifting and/or powerlifting". The study analyzes the fear of movement-related pain in athletes through a network analysis of the 40-item PHODA-SeV questionnaire. The manuscript provides an adequate review of previous studies, theoretical background and exposition of methodology and results achieved. The conclusions may be of interest to the readers of the journal and for this reason I recommend its publication after the modification of certain minors concerns that could improve the reading and comprehension of the text:

Reply: We thank the Reviewer for the positive comment and will address all comments below.

1. In the Introduction, the difference between the reflective model and network analysis is explained. However, given that the authors are committed to the network paradigm, some comparison of the results between the reflective model used in the original development of the questionnaire and the results obtained with the alternative model would be desirable in the discussion of the manuscript. (p. 5).

Reply: We thank the Reviewer for this comment. In the original manuscript, the 3rd paragraph reflects a comparison in the understanding of the PHODA with network analysis and using the individual item analysis approach. 

In terms of comparing a reflective model approach and network analysis approach, it would require the simultaneous investigation of PHODA items with other external variables. For example, Knechtle et al (2021) reported the correlation between the total PHODA score and lifting mobility to be -0.027(Knechtle et al., 2021) – with this being an example of using a reflective model approach. A network approach would be to include lifting mobility within the network analysis, to understand its correlation between individual PHODA items, after adjusting for all other relationships. Performing such as analysis is outside the scope of this paper.

2. In the Method, although a power analysis is established for the calculation of statistical power, it still seems too small a number of participants to assess the validity of a questionnaire with 40 items, even to reach a moderate effect size of 0.5. What kind of parameters were used to obtain this calculation and what software was used to perform it (p. 7).

Reply: We thank the Reviewer for this comment. We have provided a more indepth explanation of the original sample size justification L115:

The present study’s sample size was calculated based on a broader project investigating fear of movement and LBP beliefs in individuals participating in OWL and PL training. The power calculation was based on a one-sample t test where we hypothesised that participants in this cohort would have a lower PHODA-SeV total score compared to the general population. A power calculation was performed using the pwr package in R statistical software, based on a one-sample t-test. A previous study reported an improvement in the aggregate PHODA-SeV score after graded activity treatment by 10/100 [13]. Based on a standard deviation (SD) of 12/100 [8], this improvement would translate to an effect size of 0.8, which we hypothesised to be the effect size of the difference between participants in this cohort and the general population. To detect a more conservative, moderate effect size of 0.5, 64 participants were required to provide a statistical power of 0.8, at an alpha of 0.05. To account for dropouts, 70 participants were recruited.

3. I suggest to further develop the criteria for excluding 70% of the items in the network analysis, reducing it to 13 items. (p. 8).

Reply: We have added in the L156:

Moderately high collinear items were removed using a threshold ≥ 0.6 [26], which resulted in 13 items being retained for the network analysis. A threshold of 0.6 was selected as a higher threshold would have left too few items to benefit from performing a multivariate technique such as network analysis.

4. Give more information and references on how this regularized association index is calculated. I also suggest defining how to obtain the weights in gradients and color for better interpretation, for example, what indicates a positive or negative association. (p. 9).

Reply: For the information on the index, we have included the section below L171:

When estimating the network, the graphical least absolute shrinkage and selection operator (LASSO) regularization LASSO [28] was used to elicit a sparse model. If S represents the sample variance– covariance matrix, LASSO aims to estimate K by maximizing the penalised likelihood function:

log det⁡〖(K)-trace (SK)- λ ∑_(<i,j>)▒|κ_ij | 〗

A sparse model, compared to a saturated model, results in easier interpretability given that the former has comparatively fewer edges to explain the covariation structure of the data [22].The LASSO uses a tuning parameter to control the sparsity of the network, which was selected by minimizing the Extended Bayesian Information Criterion (EBIC) [29]. The graphical LASSO was run for 100 values of the λ logarithmically spaced between the maximal value of the tuning parameter at which all edges are zero (〖\\ λ〗_max), and 〖\\ λ〗_max/100. For each of these graphs, the EBIC is computed and the graph with the lowest EBIC is selected. This methodology is explained in more detail in previous tutorial papers [22, 30].

To better explain the definition of association, we have included the below L164:

A positive association between two items indicates that an increase in the magnitude of one item results in an increase in the magnitude of the second item. A negative association between two items indicates that an increase in the magnitude of one item results in a decrease in the magnitude of the second item. The thickness and colour saturation of an edge denotes the strength of the association between two nodes, which is akin to the magnitude of a Pearson correlation coefficient value.

5. Wouldn't a larger sample size be needed for the bootstraap procedure to be reliable? Please provide references to support that this method is well employed (p. 10).

Reply: We thank the Reviewer for this comment. We agree with the Reviewer that larger sample sizes are required to get more precise estimates (e.g. confidence interval) of any statistical parameters. This is true regardless if the measures of uncertainty were calculated using bootstrapping or parametrically. The greater the magnitude of uncertainty measure, the greater the sample size required to produce a more precise parameter estimate. We are unaware of ta threshold sample size to conduct bootstrapping. 

6. Give more information on how the procedure for estimating the indices by varying the sample sizes per simulation was relaized. Please explain in more detail (p. 11).

Reply: We have added the following information to this paragraph L221:

Given the network estimated, the netSimulator function of the bootnet package was used to calculate six performance indices: 1) correlation between the edge weights; 2) sensitivity – the proportion of edges present, 3) specificity – proportion of missing edges; correlation between 4) Strength, 5) Closeness, and 6) Betweenness centrality measures of the given network against that of the re-estimated network using different sample sizes [22]. The netSimulator function simulates data based on a priori network structure and edge weights, similar to how a standard statistical power simulation study simulates new data based on an hypothesised effect size being detected. Using our original network structure and edge weights, new data with varying sample sizes were simulated. Herein, we varied the sample size, n, across these values: 50, 100, 150, 200, 250, and 500. For each sample size, 2000 bootstrap samples with replacement were performed to re-estimate the network and calculate the six performance indices.

7. In Table 1, it is not clear when using means and SD since it does not agree that there are no decimals in the last four variables.

Reply: We apologise for this mistake. We have modified Table 1 and used 2 decimal places for all numeric variables.

8. Give the quantitative values of the items with more Strength. (p. 13).

Reply: We have added the values, which reads as L258:

The top three items with the greatest Strength measures were Q23 (stair ascend) at 1.00, Q8 (walking with groceries) at 0.93, and Q13 (mopping with spine flexion) at 0.90 (Figure 2, 4).

References

Knechtle, D., Schmid, S., Suter, M., Riner, F., Moschini, G., Senteler, M., . . . Meier, M.L., 2021. Fear-avoidance beliefs are associated with reduced lumbar spine flexion during object lifting in pain-free adults. Pain 162, 1621-31.

---

## [Decision Letter · Decision Letter 1]

18 Oct 2022

Pain-related fear of movement dynamics in individuals with and without low back pain participating in weightlifting and/or powerlifting training

PONE-D-22-19501R1

Dear Dr. Liew,

We’re pleased to inform you that your manuscript has been judged scientifically suitable for publication and will be formally accepted for publication once it meets all outstanding technical requirements.

Kind regards,

José M. Muyor

Academic Editor

PLOS ONE

Reviewers' comments:

Reviewer's Responses to Questions

**Comments to the Author**

1. If the authors have adequately addressed your comments raised in a previous round of review and you feel that this manuscript is now acceptable for publication, you may indicate that here to bypass the “Comments to the Author” section, enter your conflict of interest statement in the “Confidential to Editor” section, and submit your "Accept" recommendation.

Reviewer #1: All comments have been addressed

Reviewer #3: All comments have been addressed

2. Is the manuscript technically sound, and do the data support the conclusions?

Reviewer #1: Yes

Reviewer #3: Yes

3. Has the statistical analysis been performed appropriately and rigorously? 

Reviewer #1: Yes

Reviewer #3: Yes

4. Have the authors made all data underlying the findings in their manuscript fully available?

Reviewer #1: Yes

Reviewer #3: Yes

5. Is the manuscript presented in an intelligible fashion and written in standard English?

Reviewer #1: Yes

Reviewer #3: Yes

6. Review Comments to the Author

Reviewer #1: The manuscript improved compared to the previous version and the authors responded all comments. I do not have additional comments.

Reviewer #3: This manuscript reports a study designed to explore a fear of movement profile in individuals who

participate in Olympic weightlifting and powerlifting sports. All comments from reviewers 1 and 2 were properly addressed. I have no new comments for the authors to address.

7. PLOS authors have the option to publish the peer review history of their article (what does this mean?). If published, this will include your full peer review and any attached files.

Reviewer #1: No

Reviewer #3: **Yes: **Arthur de Sá Ferreira

---

## [Editor Report · Acceptance letter]

20 Oct 2022

PONE-D-22-19501R1 

Pain-related fear of movement dynamics in individuals with and without low back pain participating in weightlifting and/or powerlifting training 

Dear Dr. Liew:

I'm pleased to inform you that your manuscript has been deemed suitable for publication in PLOS ONE. Congratulations! Your manuscript is now with our production department. 

Kind regards, 

on behalf of

Dr. José M. Muyor 

Academic Editor

PLOS ONE